# Evaluation of Early Age Concrete Pavement Strength by Combined Nondestructive Tests

Katelyn Kosar [1], Lev Khazanovich [1,*] and Lucio Salles [2]

1    Department of Civil Engineering, University of Pittsburgh, Pittsburgh, PA 15261, USA
2    Department of Civil Engineering Technology, Rochester Institute of Technology, Rochester, NY 14623, USA
*    Correspondence: lev.k@pitt.edu

**Abstract:** During concrete pavement construction, comprehensive information on concrete strength development is necessary for knowledgeable scheduling decisions. To improve in situ strength estimation, nondestructive tests can be combined to maximize available information and increase collection efficiency. Ultrasonic testing has a higher accuracy in strength estimation for early age concrete and the external device allows for more data to be easily collected, while maturity testing can use temperature models to predict strength development. A procedure was created that combines these methods and their strength models in the laboratory for improved and increased field strength data. Using the procedure allows more data to be collected with greater accuracy and provides an adjustable, predicted strength development. This improves the efficiency of fast-track construction projects without resorting to costly alternatives by providing more information on strength development of in situ concrete than traditional strength testing or nondestructive methods individually.

**Keywords:** concrete pavements; nondestructive testing; ultrasonic testing; maturity; early age concrete properties

## 1. Introduction

Pavement engineers need accurate information on in situ concrete properties to make decisions on early age pavement construction procedures. When to perform procedures, such as joint sawing and opening to traffic, relies on the pavement's ability to withstand stresses applied during construction. Early age concrete properties change rapidly and there is a risk of damage if concrete is subjected to loading prematurely. Onsite concrete testing must be fast and reliable to make proper decisions on procedure timing and scheduling.

Current practices involve pouring separate specimens using the same concrete mixture in the field to be tested for compressive or flexural strength in a laboratory. This practice, called destructive testing, allows for a direct strength measurement for specimens, but is laborious, time-consuming, and may not be representative of field strength [1]. Using separate specimens ignores the effects of load, climate, edge support conditions, time of construction, design features, or other factors that may affect pavement performance but are irreplicable in a laboratory setting [2]. Variability within the pavement due to small environmental changes, concrete mixture changes, or placement differences may also be overlooked [3].

The pavement industry has recognized these limitations and has begun implementing nondestructive methods which directly inspect, monitor, or test systems without removing serviceability. Environmental, boundary, or curing conditions of in situ pavements are more likely to be accounted for in nondestructive tests [3,4]. Concrete material properties, such as strength, density, or moisture content, are estimated by empirically correlating the property to variables measured by nondestructive tests [5]. Various nondestructive tests have been performed for years to evaluate concrete pavements, often for estimating in situ concrete strength or quality control. This paper proposes the combined use of two

nondestructive methods in determining early age concrete strength development: maturity method and ultrasonic testing.

The maturity method is a popular nondestructive test that relies on the relationship between early age concrete temperature and the rate of cement hydration to estimate concrete strength. Maturity is defined as the product of concrete age and temperature or the equivalent age at a specified temperature [3,4]. When temperature is measured in the field using an imbedded sensor, the equivalent strength can be obtained through the maturity–strength relationship. An important aspect of maturity is the predictive ability of this relationship. Temperature data are predictable based on historical data on ambient temperature and heat of hydration. Since maturity is a function of temperature, temperature models can be used to predict the change in maturity and therefore the future strength development. This is a major benefit in scheduling early age construction practices and a key advantage to maturity.

The maturity method relies on a calibration between maturity index and strength. This preconstruction laboratory work must be done for each concrete mix and if any deviations occur in the field (changes in the water-to-cement ratio, aggregate, cement type, curing conditions, etc.), the relationship is no longer valid. Field use depends on temperature sensors and it is assumed the sensor location is representative of the entire pavement, which can ignore potential issues in consolidation or curing [3,4,6]. Despite limitations, maturity is an accurate and low labor method for estimating in situ concrete strength. The maturity method has been used for many years in research [3,4,7–13], has an ASTM standard for consistent use [14], and has been implemented in over half of state specifications as an acceptable strength measurement.

Ultrasonic testing allows for analyzing internal characteristics of structures using penetrating, mechanical waves. Transmitters in an ultrasonic device emit waves at a specified frequency that travel through the concrete pavement to be received again at the surface [5,15,16]. For concrete, the frequency is typically between 20 and 150 kHz, often 50 kHz. Results from ultrasonic testing provide the time it takes for a wave to travel a set distance through the concrete, which is used to determine the wave velocity. The received signal can be analyzed to estimate differences in stiffness, material properties, or defects [15]. Wave velocity is theoretically related to the elastic modulus, Poisson's ratio, and the density, which encourages a direct and accurate relationship to strength [17].

Traditional ultrasonic devices require acoustic coupling between transducers and the concrete surface that must be manually installed. Modern ultrasonic devices can be applied without coupling agents or surface preparation and have short data collection times. This allows for more freedom in scanning so variability within the pavement can be identified, critical stress locations can be monitored, or potential defects can be flagged [16]. Additionally, ultrasonic tests are unaffected by moisture or constraint conditions, unlike other nondestructive techniques, and the wave velocity–strength relationship is less sensitive to changes in the concrete mixture [17,18]. The mixed material properties that come with a heterogenous material such as concrete can affect wavelength differently and increase result variability [4]. This is addressed by using linear array systems that have numerous transducers aligned in multiple channels, so many signal time histories are used to compute wave velocity. This adds redundancies and reduces measurement variability. Linear array devices increase the repeatability of ultrasonic testing over a traditional single transducer and receiver arrangement. However, when used on small specimens, the edge conditions affect wave propagation, creating discrepancies when relating laboratory specimens to in situ pavements of the same concrete mixture and age.

Both maturity and ultrasonic testing have been used in strength estimations, but ultrasonic testing has been found to be more accurate than maturity at early ages [18–20]. The improved accuracy, in combination with the mobility and minimal set-up, make ultrasonic testing preferable for early age field use. This would remove the need for numerous permanent sensors along the pavement, as required by maturity testing.

Maturity and ultrasonic testing have been suggested for combined use in the past [5]. One study proposed ultrasonic testing with pressure waves to check if maturity was overestimating strength [21]. Another used an older form of ultrasonic testing, seismic modulus, for laboratory correlations and field use to estimate strength with success [7]. In recent years, ultrasonic testing has been upgraded and has become a popular research topic for combined use with the rebound hammer test [22–26]. This is useful but, as with ultrasonic testing on its own, it only offers a strength estimation at the time of testing. This study aims to revisit the successes of past research with modern ultrasonic technology in combination with maturity, and increase the strength development information available for present day construction. Accurate temperature modeling allows maturity to predict future strengths, but no such model exists for ultrasonic wave velocity. Combining the flexible field use of ultrasonic testing and the predicted strength development from the maturity–strength relationship provides a complete view of the concrete pavement strength.

The purpose of this paper is to combine multiple nondestructive testing procedures to increase the efficiency of concrete strength estimations and predicted development in the field. Both nondestructive tests explored in this study provide reliable strength estimations but have certain limitations that can be improved by combining procedures. This can be especially beneficial in early age concrete evaluation when high accuracy is necessary or when there is limited construction time.

To best explain the combined nondestructive procedure, this paper will give an overview of the standard practices for each nondestructive test and use a test pavement section to exemplify the process and benefits.

## 2. Materials and Methods

The test pavement was constructed on 13 July 2020, in Imperial, Pennsylvania, U.S., using a slipform paver in a work vehicle parking lot. The dowelled jointed plain concrete pavement (JPCP) consisted of 3.7 m × 3.7 m × 0.2 m (12 ft × 12 ft × 0.67 ft) concrete slabs. The concrete mixture met the requirements for Long-Life Concrete as specified by the Pennsylvania Department of Transportation (PennDOT) [27]. The ultimate compressive strength of this mixture was 37.9 MPa (5500 psi) and the ultimate flexural strength was 5.9 MPa (850 psi). Additional information on the dataset is available elsewhere [20].

Maturity testing was performed according to ASTM C1074 [14]. The temperature was monitored using thermocouples in two locations within the slab: one at the center and another near the lane edge. Data were recorded at 5-min intervals, beginning immediately after placement and continuing to 14 days. Thermocouples were also placed at the center of three cylinder and three beam specimens in a laboratory and monitored similarly to the field.

Ultrasonic testing was used to measure the shear wave velocity at six locations on each slab: one at the center, another near the lane edge, and one in each corner, taking care to avoid dowel bars. A linear array ultrasonic device, A1040 MIRA from Acoustic Control Systems, Saarbrücken, Germany, was used at a frequency of 50 kHz for wave velocity measurements. MIRA has 48 dry point contact transducers located on the bottom of the device to emit and receive signals in 12 linear array channels. The shear waves are recorded, stored, and analyzed for every scan in 66 signal time histories, and the average wave velocity over multiple measurements is reported. Data collection began approximately five hours after placement once the concrete tolerated walking on the slab surface. Measurements were collected regularly through the first day with additional testing 1, 3, 5, 7, and 14 days after placement.

Cylinder and beam specimens were cast in a laboratory using the same concrete placed. Compressive and flexural strength testing was performed using ASTM standards C39 and C78, respectively [28,29]. Three cylinder and three beam specimens were tested for strength at 1, 3, 5, 7, and 14 days. The specimens monitored for temperature were tested for strength on day 14. Before strength testing, each beam specimen was scanned on each side using MIRA for wave velocity measurements. Table 1 shows a summary of the relevant field data.

**Table 1.** Test pavement dataset.

| Test Time | Compressive Strength, MPa (psi) | Modulus of Rupture, MPa (psi) | Cylinder Maturity, °C-hr | Shear Wave Velocity, m/s |
|---|---|---|---|---|
| 1 day | 22.8 (3311) | 4.1 (597) | 847 | 2622 |
| 3 day | 29.8 (4329) | 5.1 (741) | 2006 | 2720 |
| 5 day | 30.5 (4426) | 5.0 (719) | 3192 | 2721 |
| 7 days | 34.7 (5040) | 5.6 (819) | 4409 | 2766 |
| 14 days | 36.1 (5237) | 5.7 (824) | 8604 | 2798 |

### 2.1. Maturity–Strength Relationship

The procedure to determine the maturity curve is briefly detailed below and should be conducted in accordance with ASTM C1074 [14]. Relationships are mixture-specific and separate maturity testing must be performed for different mixture designs.

The process includes monitoring concrete specimen temperatures while regularly measuring concrete strength using destructive testing. ASTM C1074 highlights five days of testing, 1, 3, 7, 14, and 28, but additional times can be added if necessary [14]. At least 15 cylinder or beam specimens are needed, so three specimens are tested at each time increment to establish the average compressive or flexural strength, respectively, for that day. Temperature sensors are placed at the center of at least two cylinder and two beam specimens. These specimens are monitored as strength testing continues on other specimens and are tested for strength on the final day. The Nurse–Saul method is commonly used in the maturity–strength relationship by computing the concrete maturity index or time-temperature factor (TTF) using laboratory temperature data and the following equation:

$$\text{TTF} = \sum (\text{T}_{\text{PCC,m}} - \text{T}_0)\Delta t, \tag{1}$$

where TTF is the temperature-time factor at age t in degree-days or degree-hours; $\Delta t$ is the time interval in days or hours; $\text{T}_{\text{PCC,m}}$ is the mean concrete temperature during the time interval $\Delta t$ in °C; and $\text{T}_0$ is the datum temperature in °C (standard values are 14 °F or −10 °C).

The temperature and strength data are correlated to create a unique maturity–strength relationship for the mixture. The following relationship between strength and maturity is assumed:

$$\text{M}_\text{r} = \text{M}_{\text{ru}}e^{-\left(\frac{a_\text{m}}{\text{TTF}}\right)^{b_\text{m}}} \tag{2}$$

$$\text{f}'_\text{c} = \text{f}'_{\text{cu}}e^{-\left(\frac{c_\text{m}}{\text{TTF}}\right)^{d_\text{m}}} \tag{3}$$

where $\text{M}_\text{r}$ is the flexural strength (modulus of rupture) in MPa (psi); $\text{M}_{\text{ru}}$ is the ultimate expected flexural strength in MPa (psi); $\text{f}'_\text{c}$ is the compressive strength in MPa (psi); $\text{f}'_{\text{cu}}$ is the ultimate expected compressive strength in MPa (psi); and $a_\text{m}$, $b_\text{m}$, $c_\text{m}$, and $d_\text{m}$ are maturity calibration coefficients.

To utilize maturity in the field, temperature sensors are placed in the pavement during pouring in a position where they are fully surrounded by concrete to monitor temperature and calculate the maturity index. The maturity of the concrete pavement is then compared in the maturity–strength relationship to determine the estimated strength of the pavement at that time.

### 2.2. Wave Velocity–Strength Relationship

In this study, the ultrasound wave propagation velocity was analyzed in relation to the concrete's Young's modulus, which strongly correlates to the concrete's flexural and compressive strength. Therefore, it can be used to estimate the strength of a given concrete mix. A linear array device called MIRA was used to determine the concrete's ultrasound velocities.

To develop the strength correlation, strength and wave velocity data must be collected from beam specimens that use a similar concrete mixture as the slab. The beam must first

be scanned using the ultrasonic device to determine the average wave velocity at that age. Immediately following the scan, the beam should be tested for flexural strength. The test section used five suggested days of testing, 1, 3, 5, 7, and 14, but additional times can be added if necessary. At least 15 beam specimens are needed, so three specimens are tested at each time increment to establish the average strength for that day. ASTM C78 and C293 are procedures that can be used to determine the flexural strength of the beams [29,30]. The following relationship between flexural strength and shear wave velocity is assumed:

$$M_r = M_{ru} \times e^{a_s SWV + b_s},$$ (4)

where SWV is the shear wave velocity in m/s and $a_s$ and $b_s$ are shear wave velocity calibration coefficients depending on the concrete mix properties.

## 3. Results

Maturity and ultrasonic testing are both reliable nondestructive methods individually but have unique advantages over each other. Combining these methodologies can use both tests to their highest ability and provide the user with thorough information on concrete strength development. This procedure will allow strength estimations from ultrasonic testing to utilize and adjust the predictive maturity–strength relationship, accurately providing both current and future field strength estimations. This proposed procedure is outlined below and is applicable using either imperial or metric units. The dataset summarized in Table 1 will be used to exemplify the procedure.

*Step 1: Performing Laboratory Testing*

As with separate maturity and ultrasonic testing procedures, preconstruction laboratory testing must be performed based on the project concrete mixture. Cylinder and beam specimens should be prepared for compressive and flexural strength testing. Several ages should be tested to establish a proper strength gain rate. The final day cylinder and beam specimens should be imbedded with thermocouples to monitor temperature for the entirety of the testing period. Each beam should be scanned with an ultrasonic device before flexural testing.

*Step 2: Establishing Maturity–Strength and Wave Velocity–Strength Relationships*

The maturity–flexural strength, maturity–compressive strength, and wave velocity–flexural strength relationships must be established using the procedures outlined in the previous sections. This will provide calibration coefficients for each of the three models.

*Step 3: Determining the Wave Velocity–Compressive Strength Relationship*

When using linear array ultrasonic devices, compressive strength specimens cannot be scanned for wave velocity. The following equation relates wave velocity to compressive strength through maturity:

$$f_c' = f_{cu}' \times \exp\left( -\left( \frac{c_m}{a_m} \right)^{d_m} (-a_s SWV - b_s)^{\frac{d_m}{b_m}} \right).$$ (5)

The required laboratory work was performed on the dataset to establish the model coefficients described in Steps 1–3 for Equations (2), (3), (4), and (5) as shown in Table 2. Visualizations of these models compared to destructive strength testing are shown in Figures 1 and 2.

**Table 2.** Strength model coefficients.

| Maturity–Strength Model | | SWV–Strength Model | |
|---|---|---|---|
| | 249.9 | $a_s$ | 0.002 |
| $b_m$ | 0.94 | $b_s$ | −5.424 |
| $c_m$ | 490.9 | | |
| $d_m$ | 1.026 | | |

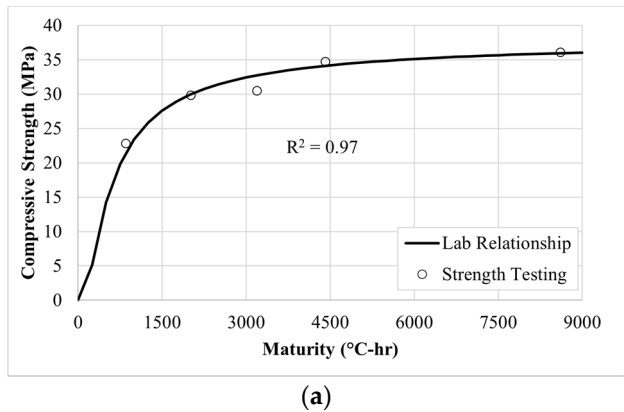
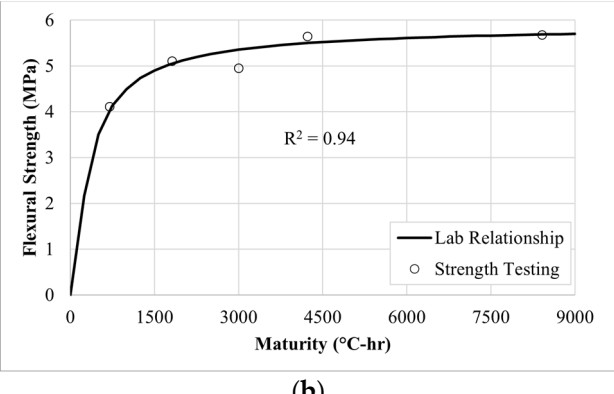

**Figure 1.** Test section maturity–strength relationships for (**a**) compressive strength and (**b**) flexural strength.

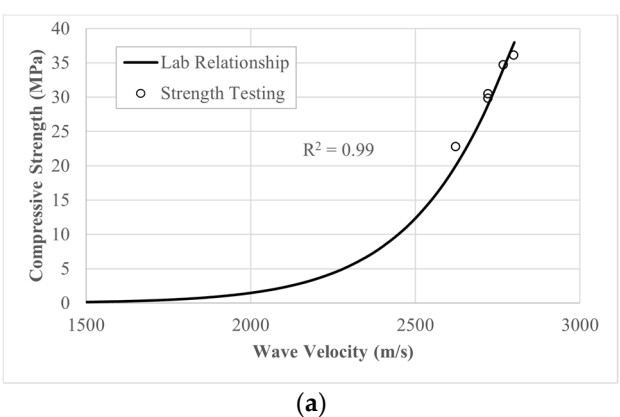
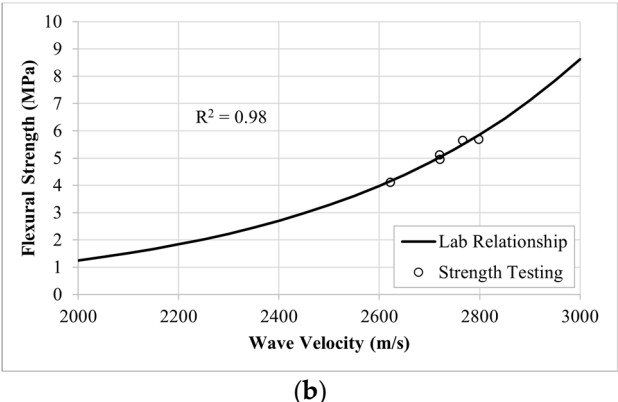

**Figure 2.** Test section SWV– strength relationships for (**a**) compressive strength and (**b**) flexural strength.

*Step 4: Choosing a Strength to Consider*

The user must select a compressive or flexural strength ($f'_{c0}$ or $M_{r0}$) to be considered for an early age procedure. This can be any strength the user may want to confirm in the field.

The example will use a compressive strength of 20.7 MPa (3000 psi). This is a common strength for opening to traffic that is typically measured with destructive testing.

*Step 5: Calculating Relative SWV*

The strength chosen in Step 4 must be converted to shear wave velocity. Equations (4) and (5) can be rearranged to find the wave velocity relative to the chosen flexural or compressive strength.

Using Equation (7) and the respective coefficients from Table 2, the relative shear wave velocity at 20.7 MPa (3000 psi) can be calculated to be 2647 m/s.

$$SWV_0 = \frac{1}{a_s} \ln \frac{M_{r0}}{M_{ru}} - \frac{b_s}{a_s} \tag{6}$$

$$SWV_0 = -\frac{1}{a_s} \left( \left( -\ln \frac{f'_{c0}}{f'_{cu}} \right)^{\frac{b_m}{d_m}} \left( \frac{a_m}{c_m} \right)^{b_m} \right) - \frac{b_s}{a_s} \tag{7}$$

*Step 6: Using SWV Field Shift Factor*

Linear array ultrasonic testing devices are affected by specimen size, which results in higher beam shear wave velocities when compared to velocities of a slab with the same concrete mixture and age. Since this procedure uses beam wave velocities to develop strength correlation factors, shear wave velocities measured on the slab estimate lower

strengths. The discrepancy between laboratory and field measurements has been identified in previous studies [19,31–33]. Salles et al. (2022) used field maturity data to adjust field velocity measurements, which resulted in an accurate correlation shift [19]. However, with this procedure, field maturity testing is no longer required, since field strength could be estimated by a more reliable ultrasonic test. Although maturity can be used to find the field shift if sensors are used in the field, an alternative shift factor is introduced.

The difference between beam and slab velocities remains relatively consistent. The ages of testing were 1, 3, 5, 7, and 14 days, and the differences between slab and beam shear wave velocities were 108, 121, 64, 107, and 99, respectively. This resulted in an average difference of 100 m/s and a standard deviation of 19 m/s. Full data such as this would be unavailable at the time this procedure is needed. Using the difference between the first field wave velocity measurement and the laboratory velocity at the same age provides a reasonable shift factor for the remaining data points (Figure 3). Zhang et al. (2020) also used the first wave velocity field measurement as a shift factor for their relationship between wave velocity and temperature, which provided accurate property predictions for the considered field material [31].

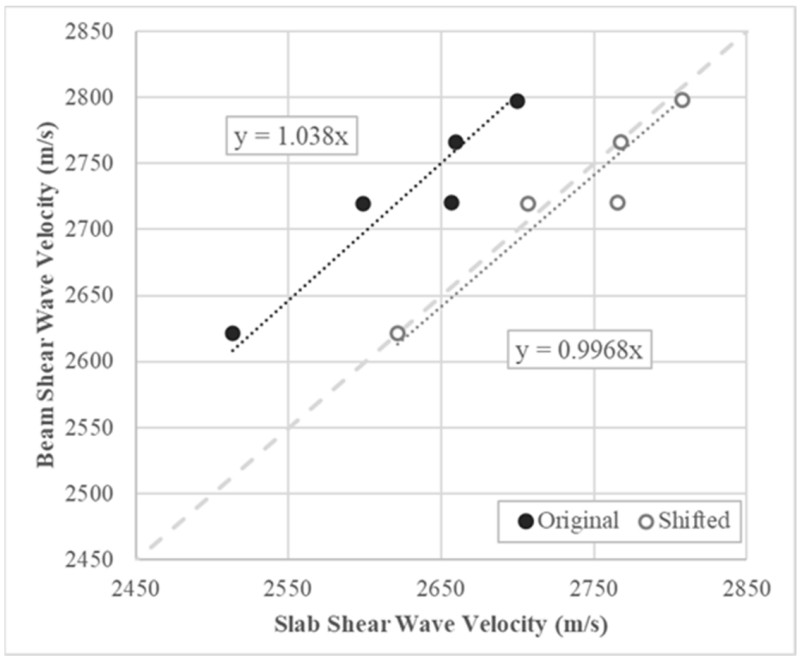

**Figure 3.** Differences between beam and slab velocities before and after shifting.

When using this procedure, wave velocity measurements should be taken at the earliest age to be tested in the laboratory. The difference between the field wave velocity and the laboratory wave velocity is determined at this age. This value can be subtracted from $SWV_0$, which allows the user to look for this velocity as it is read on the device in the field and obtain the corresponding strength.

For the test pavement, the earliest time of testing was 1 day, where the average laboratory shear wave velocity measurement was 2622 m/s. In the field, the average shear wave velocity measurement was 2513 m/s. This leads to a difference of 108 m/s to be subtracted from $SWV_0$, providing the immediate device velocity. Figure 4 shows how the field shift factor adjusts the field strength gain rate to better correspond to strengths measured in a laboratory. The dashed line represents a 1:1 ratio between beam and slab shear wave velocity and the shifted slab data is clearly closer to representing beam velocities.

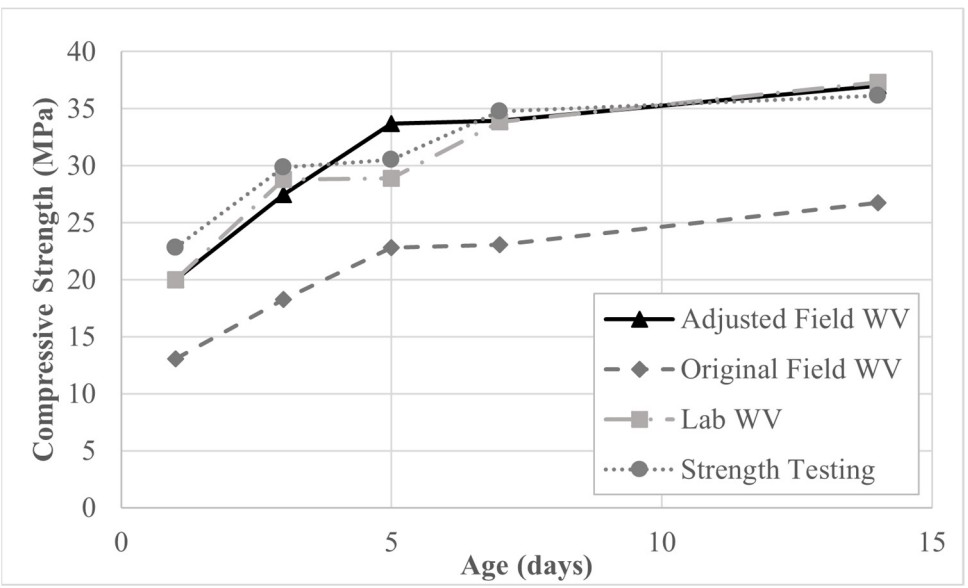

**Figure 4.** Adjusting the field wave velocity.

For the test section, the previous step determined that at a compressive strength of 20.7 MPa (3000 psi), the SWV would be 2647 m/s. Applying the field shift factor of 108 m/s, the value to be expected in the field would be 2539 m/s.

*Step 7: Calculating Relative Maturity*

The strength chosen in Step 4 must be converted to maturity. Equations (2) and (3) can be rearranged to find the maturity relative to the chosen flexural or compressive strength, respectively. Using Equation (9), the maturity for the dataset at 20.7 MPa (3000 psi) can be calculated to be 800 °C-hr.

$$\text{TTF}_0 = \frac{a_m}{\left(\ln \frac{M_{ru}}{M_r}\right)^{\frac{1}{b_m}}} \tag{8}$$

$$\text{TTF}_0 = \frac{c_m}{\left(\ln \frac{f'_{cu}}{f'_c}\right)^{\frac{1}{d_m}}} \tag{9}$$

*Step 8: Updating Concrete Strength Development for Specific Strength*

The maturity–strength relationship can be updated to display the concrete strength development after the chosen strength from Step 4 based on the anticipated change in maturity. The future strength gained after the field dataset pavement reaches 20.7 MPa (3000 psi) can now be predicted with proper temperature models:

$$M_r = M_{ru}e^{-\left(\frac{a_m}{\text{TTF}_0 + t*T_{pcc,m}}\right)^{b_m}} \tag{10}$$

$$f'_c = f'_{cu}e^{-\left(\frac{c_m}{\text{TTF}_0 + t*T_{pcc,m}}\right)^{d_m}} \tag{11}$$

where t is the time from specified strength in hours and $T_{pcc}$ is the mid-depth mean concrete slab temperature depending on the pavement location, concrete slab thickness, and construction month.

There are several methods to obtain the concrete slab mid-depth mean temperature. Thermocouples can be placed in the concrete slab and the measured temperature can be assumed for the pavement length. The measured temperature for the test pavement and resulting maturity–strength relationship can be seen in Figure 5. After the first few hours of curing, the pavement temperature relies less on heat of hydration and more on ambient temperature. The Mechanistic Empirical Pavement Design Guide uses the Enhanced Integrated Climatic Model (EICM) to estimate the hourly temperature of pavement layers

based on ambient climate data from local weather stations [34]. The average ambient temperature of Pittsburgh, PA in July 2020 was 23.8 °C. This average monthly temperature and resulting relationship can be seen in Figure 5. Despite the different temperatures, the maturity–strength relationships are similar.

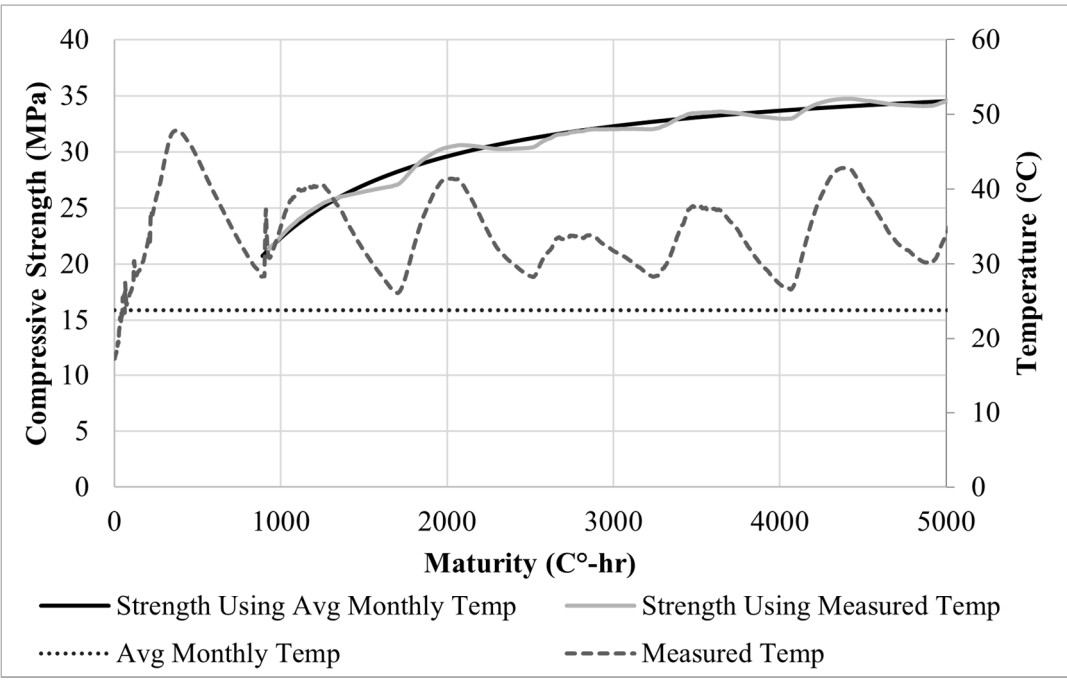

**Figure 5.** Using different temperature to determine the maturity–strength relationship after a strength of 20.7 MPa.

For the highest accuracy, temperature sensors could be used in the field, or alternatively, a temperature model based on ambient temperature, such as EICM, could be used for accurate results without the need for permanent sensor installation.

## 4. Discussion

This dataset agrees with previous research that the shear wave velocity correlates better to strength than maturity [20]. Other studies have observed inconsistencies in the accuracy of strength estimations using maturity with over-estimations [9], under-estimations [35], or both, depending on the concrete mixture and maturity–strength relationship used [19,36]. In this study, the strength estimated using maturity had an average difference from compressive and flexural strengths of 8.5% and 7.4%, respectively. Ultrasonic testing is more reliable in the field as the strength estimated using shear wave velocity had an average difference from the compressive and flexural strengths of 5.7% and 1.9%, respectively. This may appear as a minimal increase in accuracy, but when considering early age concrete, accuracy is extremely important. Early age concrete changes quickly, gaining substantial strength and stiffness in a relatively short amount of time. It is important to accurately know concrete strength to properly schedule construction procedures such as joint sawing and opening to traffic. Incorrect strengths can accelerate or delay procedures that when mistimed, can reduce the integrity of the pavement.

Another advantage of ultrasonic testing is the opportunity to minimize or eliminate the need for permanent equipment installation while increasing the monitoring area. This can be achieved by using minimal temperature sensors to obtain the mid-slab concrete temperature, or by using a temperature model such as EICM to predict the pavement temperature without any permanent installation.

Ultrasonic testing uses a single measurement of shear wave velocity at a specific time to estimate strength, while maturity can use temperature models to predict changes in

maturity and provide predictions on strength development. The relationships established in this study allow strengths estimated from ultrasonic testing to adjust the predicted strength development. By using this method repeatedly, the predicted strength gain rate can be continuously adjusted to match the more accurate strength estimate obtained from shear wave velocity. This method provides a comprehensive understanding of strength development as it evolves in the pavement.

Figure 6 shows how the strength gain rate was adjusted for the test section on days 3 and 5. The original line shows the strength gain rate from the initial pour according to the maturity–strength relationship. It aligns well with the day 1 field strength estimated by the shear wave velocity. However, on day 3, when the field strength was again determined, the prediction was overestimated. The strength gain rate can be adjusted down to predict the strength gain from that point. Again, on day 5, it was found that the prediction underestimated the strength in the field. The strength gain rate can then be adjusted up where it continued to represent the actual field strength well.

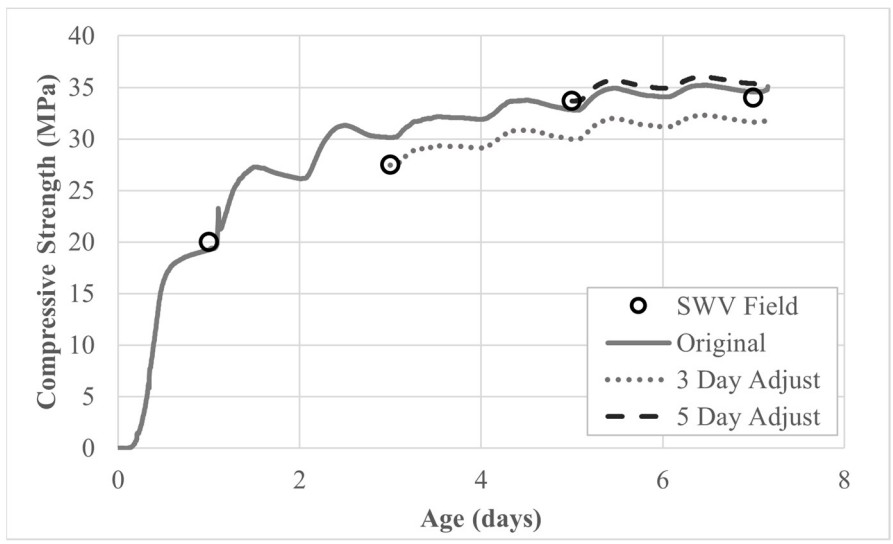

**Figure 6.** Adjusting the strength gain rate in accordance with field SWV measurements.

When considering a specific strength, the final outcomes from this procedure include the shear wave velocity to watch for in the field and the predicted strength development moving forward in construction. This collaboration of nondestructive tests increases the available concrete strength information for immediate field use and for future construction scheduling.

## 5. Conclusions

Early age strength is a critical concrete property for pavement engineers, who rely on accurate strength information to schedule time-sensitive construction procedures. Any additional data gathered quickly and reliably allows them to make more informed decisions on construction scheduling. Combining nondestructive testing is recommended to improve available strength data. The process outlined in this paper combines two popular techniques, each with unique advantages. After applying the procedure outlined above, the nondestructive tests can be utilized together to determine strength estimations and development for more comprehensive results in the field. This combination also allows the predicted strength development to be modified in the field to best represent the in situ strength, as measured by ultrasonic testing. The benefits of this procedure are most evident during construction because of the flexibility in testing. Ultrasonic testing can be used quickly and often at any point of concern on the pavement to determine the strength and strength variability throughout construction. Thermal sensors or temperature modeling can then be used to determine future strength development to estimate when construction procedures, such as opening to traffic or joint cutting, can be performed. It is recommended

to perform this procedure on rehabilitation projects or projects that require fast construction because it encourages efficient construction that reduces time without increasing costs or changing common practices.

**Author Contributions:** Conceptualization, L.K., K.K. and L.S.; data collection, L.K., K.K. and L.S.; methodology development, L.K. and K.K; draft manuscript preparation, K.K. and L.K.; All authors have read and agreed to the published version of the manuscript.

**Funding:** This research was supported by the University of Pittsburgh Center for Impactful Resilient Infrastructure Science and Engineering (IRISE) and the University of Pittsburgh Anthony Gill Chair.

**Institutional Review Board Statement:** Not applicable.

**Informed Consent Statement:** Not applicable.

**Data Availability Statement:** Not applicable.

**Acknowledgments:** The authors would like to thank Golden Triangle Construction for allowing the testing in their facilities and the FHWA Concrete Mobile Laboratory for providing the ultrasonic tomography device, MIRA, and maturity meter.

**Conflicts of Interest:** The authors declare no conflict of interest.

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
