# Peer review of "Evaluation of Early Age Concrete Pavement Strength by Combined Nondestructive Tests"

_applsci, doi:10.3390/app13042240_

Round 1
Reviewer 1 Report
The article presents an interesting research on the use of complementary nondestructive tests for the evaluation of early age concrete pavement strength. The manuscript is very valuable foe research and engineering community. I have only few minor comments before further processing:
- Please consider to use the word “complementary” instead of “combined” in the title as well as in the whole article,
- Please highlight more deeply the novelty of the article in the abstract and in the end of the introduction,
- Please delete numbers (1, 2, 3 etc) in keywords as it is not necessary to number this section,
- Please do not use citation pockets (e.g. [2, 3] or [7-13]) but rather cite each reference individually showing the reason why each reference has been cited in the article (e.g. as pointed out by the authors of the article [8]…). Please remove unnecessary references that can not been cited in this way,
- I would like to see some perspectives in the conclusions section.
Author Response
The article presents an interesting research on the use of complementary nondestructive tests for the evaluation of early age concrete pavement strength. The manuscript is very valuable for research and engineering community. I have only few minor comments before further processing:
- Thank you for the comments.
Please consider to use the word “complementary” instead of “combined” in the title as well as in the whole article,
-
- Respectfully, we believe that “Combined” is more representative in this case since the majority of the procedure outlined in this paper is combining laboratory procedures and models. Ultrasonic velocity can then be used alone or complementary to maturity in the field. Adjustments were made in the body of the paper to clarify.
- Please highlight more deeply the novelty of the article in the abstract and in the end of the introduction,
- A larger emphasis on the paper’s novelty was added to several sections of the paper. We hope that the edits add clarification.
- Please delete numbers (1, 2, 3 etc) in keywords as it is not necessary to number this section,
- Formatting was corrected.
- Please do not use citation pockets (e.g. [2, 3] or [7-13]) but rather cite each reference individually showing the reason why each reference has been cited in the article (e.g. as pointed out by the authors of the article [8]…). Please remove unnecessary references that can not been cited in this way
- We are using a conventional form of citation to express consensus or abundancy of studies in respect to an idea. Many of the citations in this format reference similar techniques or conclusions. Addressing each separately would result in a lot of unnecessary repetition. Each case was considered individually and when necessary, citations were separated or removed.
- I would like to see some perspectives in the conclusions section.
- Recommendations about when this procedure will be most beneficial were added to the conclusions.
Reviewer 2 Report
The manuscript discusses a proposal to combine nondestructive tests to predict concrete strength in the field. The paper needs improvement, consider my comments for improvement.

Author Response
The manuscript discusses a proposal to combine nondestructive tests to predict concrete strength in the field. The paper needs improvement, consider my comments for improvement.
- Thank you for your detailed review.
- what is the novelty of this work? Similar work has been carried out by many researchers.
- These specific tests have been found to work well together in the past but now the technology of ultrasonic testing has advanced to the point that it is more practical to use in the field. This has been recognized in recent studies where ultrasonic testing is used with the rebound hammer test but this limits strength data to the time of testing. By combining ultrasonic testing with maturity, the strength development can be predicted to improve construction scheduling. Edits have been made to clarify this novelty and we appreciate you identifying the issue.
- add key specific results in abstract to give reader flavour of work
- The abstract was adjusted to specify the benefits and results from using the proposed procedure.
- remove numbering
- Formatting was corrected.
- introduction part is too long. Make it concise.
- The introduction was edited to be more direct and shortened.
- it is a factor of strength, it will always have a good correlation
- Agreed, maturity correlation is a reliable method of strength estimation.
- is this the only device that can be used? why MIRA was used? what is so special about it?
- MIRA is not the only ultrasonic testing device available, but it was the option available for this study. The introduction to linear array devices has been generalized and the device specifics have been moved to methods.
- what is the location of sensor? once again location of sensors is critical
- The sensor location within the specimen and for field use has been specified to reflect the ASTM standard.
- the curves are low quality. The data presented is not enough for a correlation
- Data amount is appropriate for testing for determining early age concrete strength in the field. The strength data shown is the average of three measurements per day of testing. Picture quality was improved.
- show correlation coefficients
- Correlation coefficients were added to the figures
- relate your results with existing literature
- A summary of results was added to the discussion section.
- summarize your work and relate it with objectives set out
- The conclusion has been edited to better summarize the paper.
- grammar errors
- Corrected, thank you for pointing them out.
Reviewer 3 Report
This manuscript evaluates the “Evaluation of Early Age Concrete Pavement Strength by Combined Nondestructive Tests”. The manuscript is elaborately described and contextualized with the help of previous and present theoretical background and empirical research. All the references cited are relevant to this area of research and also adequate. The methods/analytical study are clearly stated. The result and discussion of the research are coherent and balanced. The conclusions are supported by the results. However, some minor corrections need to be addressed before the acceptance the Manuscript.
1. Abstract: Include the specific results of this research in the abstract. Also, Mention the research recommendation of your work
2. Keywords: Remove the numerals 1, 2 etc.
3. The line ‘This practice, called destructive testing, allows for a direct strength measurement for specimens but is laborious, time-consuming, and may not be representative of field strength’ should require more citation. You can use the following works
https://doi.org/10.1002/suco.201800355
https://doi.org/10.17533/10.17533/udea.redin.20190403
4. How your work differs from the past studies?
5. Mention the novelty of your work.
6. Show the experimental photos in your manuscript.
7.Fig. 2,4. Provide clear image.
8. Discussion section ned to be strengthened by comparing your results with past researches.
9. In conclusion, mention your research recommendation at last.
10. Use more recent research for citing your work.
Author Response
This manuscript evaluates the “Evaluation of Early Age Concrete Pavement Strength by Combined Nondestructive Tests”. The manuscript is elaborately described and contextualized with the help of previous and present theoretical background and empirical research. All the references cited are relevant to this area of research and also adequate. The methods/analytical study are clearly stated. The result and discussion of the research are coherent and balanced. The conclusions are supported by the results. However, some minor corrections need to be addressed before the acceptance the Manuscript.
- Thank you for your detailed review.
- Abstract: Include the specific results of this research in the abstract. Also, Mention the research recommendation of your work
- The abstract was adjusted to specify the benefits and results of using the proposed procedure. The recommendations added to the conclusions were added to the abstract.
- Keywords: Remove the numerals 1, 2 etc.
- Formatting was corrected.
- The line ‘This practice, called destructive testing, allows for a direct strength measurement for specimens but is laborious, time-consuming, and may not be representative of field strength’ should require more citation. You can use the following works
- https://doi.org/10.1002/suco.201800355
- https://doi.org/10.17533/10.17533/udea.redin.20190403
- We included additional references to that sentence.
- How your work differs from the past studies? Mention the novelty of your work.
- A larger emphasis on the paper’s novelty was added to several sections of the paper. These specific tests have been found to work well together in the past but now the technology of ultrasonic testing has advanced to the point that it is more practical to use in the field. This has been recognized in recent studies where ultrasonic testing is used with the rebound hammer test but this limits strength data to the time of testing. By combining with maturity, the strength development can be predicted to improve construction scheduling. Edits have been made to clarify this novelty and we appreciate you identifying the issue.
- Show the experimental photos in your manuscript
- Thank you for the recommendation but since this paper is focused more on the procedure, not the experimental data we decided not to include actual photos of data collection. The paper that focused more on that dataset specifically is referenced if the reader would like more information. That paper does include experimental photos.
- 2,4. Provide clear image.
- Picture quality has been improved for all figures.
- Discussion section needs to be strengthened by comparing your results with past researches.
- A summary of results was added to the discussion section.
- In conclusion, mention your research recommendation at last.
- Recommendations about when this procedure will be most beneficial were added to the conclusions.
- Use more recent research for citing your work.
- More recent sources have been added. Thank you for pointing that out.
Reviewer 4 Report
Some comments which greatly enhance the understanding of the paper and its value are presented below. Specific issues that require further consideration are:
- The title of the manuscript is matched to its content.
- The Introduction generally covers the cases.
- The methodology was clearly presented.
- In the Reviewer’s opinion, the current state of knowledge relating to the manuscript topic has been presented, but the author's contribution and novelty are not enough emphasized.
- Experimental program and results looks interesting and was clearly presented.
- In the Reviewer’s opinion, the bibliography is representative.
- An analysis of the manuscript content and the References shows that the manuscript under review constitutes a summary of the Author(s) achievements in the field.
In the Reviewer’s opinion the manuscript is well written, and it should be published in the journal after minor revision.
Author Response
Some comments which greatly enhance the understanding of the paper and its value are presented below. Specific issues that require further consideration are:
- The title of the manuscript is matched to its content.
- The Introduction generally covers the cases.
- The methodology was clearly presented.
- In the Reviewer’s opinion, the current state of knowledge relating to the manuscript topic has been presented, but the author's contribution and novelty are not enough emphasized.
- Experimental program and results looks interesting and was clearly presented.
- In the Reviewer’s opinion, the bibliography is representative.
- An analysis of the manuscript content and the References shows that the manuscript under review constitutes a summary of the Author(s) achievements in the field.
In the Reviewer’s opinion the manuscript is well written, and it should be published in the journal after minor revision.
Thank you for your review. We appreciate your comments, and it seems that your primary concern is the novelty was not emphasized. Edits were made throughout the paper to address this concern.